# Six Weeks of Calorie Restriction Improves Body Composition and Lipid Profile in Obese and Overweight Former Athletes

**DOI:** 10.3390/nu11071461

**Published:** 2019-06-27

**Authors:** Joanna Hołowko, Małgorzata Magdalena Michalczyk, Adam Zając, Maja Czerwińska-Rogowska, Karina Ryterska, Marcin Banaszczak, Karolina Jakubczyk, Ewa Stachowska

**Affiliations:** 1Department of Biochemistry and Human Nutrition, Pomeranian Medical University, 71-460 Szczecin, Poland; 2Department of Sport Nutrition, The Jerzy Kukuczka Academy of Physical Education in Katowice, 40-065 Katowice, Poland

**Keywords:** calorie restriction diet, body mass reduction, insulin, IGF-1, leptin, adiponectin

## Abstract

Objective: The aim of the study was to compare the impact of 6 weeks of reducing daily caloric intake by 20% of total daily energy expenditure (TDEE)-CRI vs. reducing daily caloric intake by 30% of TDEE-CRII on body mass reduction and insulin metabolism in former athletes. Methods: 94 males aged 35.7 ± 5.3 years, height 180.5 ± 4.1 cm, and body mass 96.82 ± 6.2 kg were randomly assigned to the CRI (*n* = 49) or CRII (*n* = 45) group. Thirty-one participants (18 subjects from CRI and 13 from CRII) resigned from the study. The effects of both diets on the body composition variables (body mass—BM; body fat—BF; fat free mass—FFM; muscle mass—MM; total body water—TBW), lipid profile (total lipids—TL; total cholesterol—TCh; HDL cholesterol—HDL; LDL cholesterol—LDL; triglycerides—TG), and glucose control variables (glucose—GL, insulin—I, HOMA-IR, insulin-like growth factor-1—IGF-1, leptin and adiponectin) were measured. Results: After adhering to the CR I diet, significant differences were observed in FFM, MM and TG. After adhering to the CR II diet, significant differences were registered in tCh, TL and LDL. Both diets had a significant influence on leptin and adiponectin concentrations. Significant differences in FFM, MM, and tCh were observed between the CR I and CR II groups. At the end of the dietary intervention, significant differences in BF, FFM, MM and TBW were observed between the CR I and CR II groups. Conclusion: The 6 weeks of CR II diet appeared to be more effective in reducing BF and lipid profile and proved to be especially suitable for subjects with high body fat content and an elevated level of lipoproteins and cholesterol. Both reductive diets were effective in improving the levels of leptin and adiponectin in obese former athletes.

## 1. Introduction

In many sports disciplines, and especially those in which muscular strength and body mass are important for performance, lack of physical activity and diet modification after the end of the career cause a drastic increase in body mass [1]. This often leads former athletes to become overweight or even obese [1,2,3]. In spite of the fact that this phenomenon appears to be frequent, the problem of excessive weight and obesity has not been studied extensively among former athletes, as their health problems remain outside of the mainstream of scientific interest [1]. Recently, only a few papers have been published regarding this topic [1,2,3,4,5], indicating that the maintenance of normal body mass in former athletes is a serious health issue [6]. Weight gain in this population is a consequence of three parallel processes: a decrease in the basal metabolic rate, the reduction of daily energy expenditure, and the excessive consumption of food [3,4]. The reduction in training intensity leads to a loss of muscle mass, which, in turn, determines the value of the necessary daily caloric intake. Former athletes are often accustomed to excessive calorie intake, which causes gains in body mass [4]. At the same time, a significant reduction in the volume and intensity of daily physical activity is observed, and as a consequence, a decreased resting metabolic rate occurs [7].

Currently, there are no clearly defined standards of nutrition for athletes terminating their sports careers [1,3]. It appears that the metabolic changes caused by intense exercise and nutrition specific for particular sports require a different approach to the diet of both the current and former athletes [6]. It is essential to consider that there are sport disciplines that require enormous energy consumption and expenditure during training and competition, as well as sports that prefer calorie restriction and low body mass [8]. In aesthetic sports disciplines that concentrate on movement coordination and physique (e.g., artistic gymnastics), calorie restriction and body mass control are key factors of peak performance [9]. Thus, a correctly adjusted diet for a standard subject may not be adequate for former athletes, justifying the search for more efficient dietary management of athletes at the end of their sports careers. A logical solution for former athletes with increased body mass and fat content includes substantial calorie restrictions (CRs) [1]. The available studies confirm the positive influence of CR diets on body mass reduction, blood pressure, variables of glucose metabolism, lipid profile, and immune response [1,10,11,12]. Several studies have confirmed that reducing the amount of calories in the diet by approximately 30% can lead to significant health benefits [13,14]. 

Besides poor eating habits, especially excess calorie intake, leptin and adiponectin are important variables that influence excessive weight and obesity. Therefore, they have drawn the attention of scientists engaged in metabolism and obesity studies. Leptin and adiponectin independently and inversely influence such phenomena as the insulin resistance of tissues, glucose metabolism, and vessel inflammation [15]. It has been proven that leptin can be associated with the complications resulting from obesity, such as hypertension and cardiovascular diseases [16]. Adiponectin has an inverse effect to leptin. This cytokine has an anti-atherosclerotic effect; it increases insulin sensitivity [17] and has anti-inflammatory properties [18]. The concentration of adiponectin in plasma is inversely proportional to the body mass index (BMI), and the concentration of insulin and triglycerides is directly proportional to the concentration of HDL [19].

Apart from the mentioned hormones, insulin-like growth factor (IGF-1) also plays a key role in glucose and lipids metabolism [20]. IGF-1 administration has been shown to reduce serum glucose levels in healthy individuals but also in insulin-resistant patients [20]. IGF-1 promotes fatty acid transport in muscle and its inhibition causes severe consequences like insulin resistance and even diabetes [20,21]. A deficiency of IGF-1 in adults is associated with impaired muscle mass, bone density, and lipid levels [20]. On the other hand, increased levels of free IGF-1 are observed in obese subjects [22]. 

Excess body fat is a risk factor for insulin resistance [1,23,24]. There are a number of well-established tests used to measure insulin resistance (IR) [25], including the homeostasis model assessment (HOMA). The HOMA-IR model is a simple, noninvasive method for predicting insulin resistance in middle-aged people with proper glucose tolerance [25]. This mathematical model is based on the reciprocal loop theory between the liver and β-cells of the pancreas, which regulate the concentration of glucose and insulin. The model can be used to evaluate the function of the β-cells of the pancreas as well as the level of insulin resistance [21]. To determine the HOMA index, fasting glucose and insulin values must be available. A higher level of HOMA-IR is observed in overweight, obese subjects but also in former athletes with body mass imbalance [1,23,24]. 

Overweight and obese subjects are prone to metabolic diseases such as hypercholesterolemia or hyperglycemia [1]. Thus, it is of great importance to monitor the above-mentioned blood variables in these patients under calorie-restriction diets [1]. It is expected that reduced calorie intake would decrease the concentration of blood glucose and insulin, while at the same time improving the lipid profile, decreasing the risk of diabetes [25]. A calorie restriction diet should also regulate the level of adiponectin, which has a favorable effect on the cardiovascular system. 

The aim of this study was to determine whether a dietary intervention based on the introduction of two different types of calorie restriction induce significant changes in body mass and body composition, as well as metabolic indicators, such as insulin resistance (HOMA-IR), concentration of leptin, adiponectin, IGF-1, glucose, total cholesterol, triglycerides, and HDL and LDL cholesterol. The first reductive diet decreased total daily energy expenditure (TDEE) by 20% (CR I), while the second one was based on a 30% reduction of TDEE (CR II).

## 2. Materials and Methods

### 2.1. Subjects

Ninety-four (94) Caucasian males were initially qualified for the study. All of the qualified individuals were obese former athletes as determined by the body mass index (BMI ≥ 30). The study subjects included former athletes recruited from the following sport disciplines: canoeing, rowing, swimming, athletics, soccer, as well as weightlifting and powerlifting. During their career, which lasted 10 years on average, each subject trained intensively from 6 to 10 times per week. The time period since the end of their careers did not exceed 5 years. Volunteers were randomly assigned to one of the two intervention groups CRI: *n* = 49 and CRII *n* = 45. For randomization, we used the sealed envelopes method. Specifically, during the first visit to the laboratory, each participant drew an envelope with the prescribed diet. The participants who were classified into the CRI group, consumed daily 20% calories less than the total daily energy expenditure (TDEE), whereas the participants assigned to CRII group reduced their daily calorie intake by 30% of TDEE (Table 1). Thirty-one participants resigned from the study (18 subjects from CRI and 13 from CRII) (Figure 1). These individuals were not able maintain the calorie-restricted diets and consumed fast foods, sweets, and alcohol, which were not included in the prescribed diets. Of all the subjects recruited for the study, 31 from CRI and 32 from the CRII group completed the experiment. After conducting the experiment, the study participants were further divided into three subgroups (Figure 1): CRI—1.5–2.5 kg, *n* = 13/2.5–3.0 kg, *n* = 10/above 3.0 kg, *n* = 8; CRII—1.5–2.5 kg, *n* = 14/2.5–3.0 kg, *n* = 11/above 3.0 kg, *n* = 7, depending on the range of body mass reduction.

During the initial visit, the volunteers were advised by a dietician to maintain their habitual lifestyle and current physical activity. The exclusion criteria were as follows: The intake of any supplements with established antioxidant properties; energy expenditure of physical activity > 3000 kcal/week; hypercholesterolemia (total cholesterol > 8.0 mM or dyslipidemia therapy); diabetes (glucose > 126 mg/dL or diabetes treatment); hypertension (systolic blood pressure >140 mmHg and/or diastolic blood pressure > 90 mmHg or antihypertensive treatment); multiple allergies; celiac disease or other intestinal diseases; any condition that could limit the mobility of the subject, making laboratory visits difficult; life-threatening diseases or conditions which could worsen adherence to the measurements or treatments; vegetarianism or the need for other specific diets; and alcoholism or other drug addiction. Written informed consent was obtained from all participants. The study protocol was approved by the ethics committee of the Pomeranian Medical University in Szczecin, Poland (ethic reference KB-0012/53/11) and conformed to the ethical guidelines of the 1975 Declaration of Helsinki.

### 2.2. Dietary Intervention

The dietary intervention lasted 6 weeks. The first group of participants received a mix diet with a calorie reduction of 20% of the TDEE-CRI, while the second group was prescribed a similar diet with a calorie reduction of 30% of TDEE-CRII. TDEE was calculated according to the commonly accepted model (TDEE = RMR × AF) [26]. Resting metabolic rate (RMR) was measured during the first visit and during the subsequent control visit by means of the Fitmate apparatus (Pro, COSMED, Rome, Italy). The precise Cosmed Fitmate system was used to determine the RMR energy expenditure. The whole test took approximately 20 min. The device uses the calorimetry method and directly measures the amount of oxygen uptake (with accuracy of ± 0.02%) which makes it possible to measure resting energy expenditure (REE), the resting metabolic rate (RMR) and the basic metabolic rate (BMR). The Canopy measuring cap was used for the measurement. The activity factor (AF) was determined based on the available indicator—2.0-high activity/1.6-medium/1.4-low/and 1.2-sedentary lifestyle [26]. In order to establish the AF, each participant during the first visit was administered a validated International Physical Activity Questionnaire—IPAQ. This questionnaire refers to the work-related PA as well as activities performed at home and its surroundings, any kinds of activities performed in free time (in relation to moderate or intensive physical activity) and time spent sitting. The CRI and CRII diet model provided a total daily intake of 2192 ± 54 kcal and 2133 ± 42 kcal, respectively. The following proportions of carbohydrates, fats and protein were prescribed for both study groups: CRI—3 g/kg/body mass of carbohydrates, 0.8 g/kg/body mass of fats and approximately 1.2 g/kg/body mass of protein. In the CRII diet, the participants consumed 2.9 g/kg/body mass of carbohydrates, 0.74 g/kg/body mass of fats and 1.1 g/kg/body mass of protein. The composition of the diets is presented in Table 1. The applied macronutrient composition was recommended according to the actual norms of nutrition for the Polish population as published by the Food and Nutrition Institute [27]. The meals were prepared in the form of 24-h menus for 7 days of the week. They also consisted of instructions for five meals per day. The particular diet composition was analyzed using DIETETYK 6.0 software (Jumar, Poland). The scheme of the experimental protocol is shown in Figure 1.

### 2.3. Diet Control

After 3 and 6 weeks of the experiment, the prescribed diets were checked for quantity and quality. The participants were asked to complete a 72 h food diary (2 weekdays and 1 weekend day). To increase the accuracy of the recorded data, each participant received a diet diary booklet that contained menus, pages to record foods, and photographs of food that depicted portion choices for a common food item. The dietician indicated that each study participant should record the food brand and portion size. The amounts consumed were recorded in household units, by volume or by measuring with a ruler. In addition, each subject was interviewed about their dietary patterns in the previous 6 weeks. The dietary records were validated by a nutritionist. Diets were reconstructed from diary entries received from participants in the Diet 5 program and referred to the currently valid standards recommended by the National Food and Nutrition Institute in Warsaw. Nutrient analyses were carried out using the corresponding Polish food table [27]. The participants who did not report to the control visit at the laboratory, as well as those who did not reach the recommended calorie values programed in the experiment due to the excessive consumption of sweetened beverages or alcohol, were eliminated from the study.

### 2.4. Anthropometric Data

The evaluations of body composition (body mass—BM, kg; body fat BF, %; fat free mass—FFM, %; muscle mass—MM, %; total body water—TBW, %) were carried out twice using electric impedance multi-frequency measurement BIA-1, Akern, Bioresearch SRL, PONASSIEVE, Florence, Italy [28,29]. Based on the obtained values, BMI was calculated using the following formula (BMI = body mass (kg)/height[m]^2^). The anthropometric data of former athletes before the beginning of the study is summarized in Table 2.

### 2.5. Biochemical Variables and Test Procedures

The following blood metabolic variables were evaluated in all study participants: glucose (GL), total cholesterol (TCh), HDL cholesterol (HDL), LDL cholesterol (LDL), triglycerides (TG), total lipids, insulin-like growth factor-1 (IGF-1), leptin, adiponectin, HOMA-IR, and insulin (I). Fasting blood samples were collected between 08:00 and 10:00 am. at the beginning of the study and after 6 weeks of intervention. After an overnight fast, venous blood samples for lipid analyses were collected into tubes with EDTA anticoagulant, and for glucose estimation in tubes with sodium fluoride and for hormones and IgF-1 in serum tubes. The blood was immediately placed on ice or in a refrigerator, and samples were centrifuged at 2000× *g* (2500 rpm) for 10 min at 4 °C within 2 h of collection. The plasma was then immediately stored in conditions to minimize artificial oxidation (i.e., with an antioxidant cocktail under an inert atmosphere). Standard blood biochemical analyses such as blood lipid profile and glucose concentration were carried out at the University Hospital Laboratory. The ELISA-automated microparticle enzyme immunoassay test kits for quantitative assessments of leptin, adiponectin and IGF-1 (all from R & D Systems, Minneapolis, MN, USA) were employed. HOMA-IR was calculated from the formula—fasting insulin level × fasting glucose divided by 22.5 [30].

### 2.6. Statistical Analysis

The Shapiro-Wilk, Levene, and Mauchly’s tests were used in order to verify the normality, homogeneity, and sphericity of the sample’s data variances, respectively. The verification of the differences between the analyzed values before and after the diet intervention as well as between CRI and CRII diets were verified using ANOVA with repeated measures. Statistical significance was set at *p* < 0.05. Effect sizes (η^2^ Eta-squared) were reported for the results where appropriate (Table 3). Parametric effect sizes were defined as large for η^2^ > 0.14, as moderate for 0.06 and small for 0.01 [31,32]. The statistical significance was set at *p* < 0.05. All statistical analyses were performed using Statistica 9.1 and the results were presented as means with standard deviations. 

## 3. Results

Of the 96 randomly chosen participants, 63 completed the 6-week study (Figure 1). No significant differences were noted between CR I and CR II groups for all measured variables before and after the diet intervention, except tCh and LDL (Table 4). After the diet intervention, a statistically significant decrease in TG, TL and leptin, and an increase in adiponectin levels were observed in both the CR I and the CR II group (Table 4). A high value of SD in BM before and after the CRI and CRII diet (Table 4), resulted in the groups being divided into subgroups, depending on the body mass reduction. It was revealed that after 6 weeks of diet in the CRI 1.5–2.5 subgroup, FFM (kg) and MM (%) were significantly reduced (Table 5). There were significant differences in BF, FFM (%), MM, TBW (Table 5), and tCh (Table 5) between CRI and CRII in the 3 kg subgroup. After the 6-week intervention, in the CRI 1.5–2.5 subgroup there was a significant TG decrease (Table 6). In addition, after the same amount of time in the three subgroups of CR II, a significant decrease was observed in the tCh TL and LDL level (Table 6). Both CR I and CR II induced significant increases in adiponectin concentration in all subgroups, while leptin concentration was significantly reduced in all subgroups after the CR I and CR II diet model (Table 6). After the diet intervention, a statistically significant decrease in leptin and an increase in adiponectin levels were observed in all subgroups in both the CR I and CR II diet (Table 6). 

## 4. Discussion

The current literature does not provide scientifically-based nutritional standards for former athletes [33,34,35]. However, there are studies concerning nutritional methods and observations of the health status of these subjects [5,36,37,38]. In the present study, from the 96 randomly chosen participants to consume caloric-restriction diets, 63 completed the 6-week study. Thirty-one participants (18 subjects from CR I group and 13 subjects from CR II) resigned from the study. The high number of participants who discontinued the study was caused by many diet restriction such as the need to have regular meals or no snacking. When participants met with the dietitians, they which informed them that despite their initial readiness, they could not resign from their usual diet habits especially from snacking, eating sweets or consuming alcohols. Unfortunately, it is a typical problem, psychological rather than physiological, common for obese and overweight individuals. Although these people are aware of the fact that high body mass and fat content significantly increase the risk of developing many fatal disorders, they cannot commit fully to the dietary restrictions. Studies show that individuals who decide to reduce their body mass and fat content, apart from dietetics, require psychological care [39,40].

In the present study, both CR I (reducing daily caloric intake by 20% of TDEE) and CRII diets (reducing daily caloric intake by 30% of TDEE), when adjusted to the caloric needs of a participant, helped to reduce body mass, thus improving BMI. In former athletes who applied the reducing daily caloric intake by 30% of TDEE and lowered their body mass by 1.5–2.5 kg, 2.5–3.0 kg, and over 3.0 kg, a significant improvement in lipid variables (tCh, LDL and TL) was observed, as insulin and levels were decreased and HOMA-IR was reduced. When comparing the CRI and CRII groups, it appeared that a more drastic reduction of calories observed in the CRII model improved the variables of lipid metabolism to a greater extent. This was especially visible in subjects who reduced their body mass by more than 3 kg. In former athletes who reduced daily caloric intake by 20% of TDEE and achieved a body mass loss of 1.5–2.5 kg, a significant decrease in FFM and MM was observed. This suggests that a calorie restriction diet does not protect the muscle tissue from being catabolized [41]. The lower glucose concentration resulting from this calorie restriction can be explained by a greater uptake of glucose by increased muscle mass. 

Losing weight and increasing physical activity independently has a beneficial effect on the glucose metabolism and insulin sensitivity [42]. Improvement in insulin sensitivity with dietary intervention depends on the number of calories reduced. The more restrictive the diet, the greater improvement in insulin sensitivity [42]. Our study participants demonstrated no significant improvement in glucose, I or HOMA- IR, as they consumed a relatively high number of calories daily (2192 ± 51 kcal). Insulin sensitivity improvement can be observed in healthy subjects even in short-lasting but very low-calorie diets, where calorie intake can only be 500 kcal/day [43]. Such a diet model induces a marked decrease in abdominal adipo tissue, reduction in adipocyte size, an increased adipo tissue gene expression of mitochondrial biogenesis markers and non-mitochondrial oxygen consumption pathways, as well as improved whole-body insulin sensitivity [43]. Other authors confirm that calorie restriction helps decrease body mass, reduces insulin, glucose and HOMA-IR, and stabilizes their level, even if the subjects are not physically active [1,10,12]. Research on reducing daily caloric intake by 30% of TDEE has also been carried out on subjects with type 2 diabetes. The results indicate that this relatively mild caloric restriction helps decrease the level of insulin in diabetic patients [44]. Weiss and colleagues [45] in their research performed on 52 men and women aged 45–65 observed a decrease in the level of insulin and glucose and an improvement in insulin sensitivity after the introduction of a diet based on calorie restriction. 

The loss of body mass due to calorie restriction significantly affects leptin and adiponectin secretion in fat tissue [46]. Indeed, strong evidence indicates that lowering body mass contributes to the reduction of blood leptin concentration. In the research carried out by Wing et al. [47] on 52 obese women, after the introduction of a calorie-restricted diet, the average decrease of body mass was 8.1 kg and the concentration of leptin was lowered from 30.1 to 20.4 ng/mL [47]. Sartorio et al. [48] in a study that lasted 3 weeks and involved a group of 54 obese patients, achieved a statistically significant reduction in plasma leptin concentration in both men and women (women: from 41.1 ± 3.6 ng/mL to 29.9 ± 3.0 ng/mL; men: from 19.4 ± 2.6 ng/mL to 11.6 ± 1.3 ng/mL) [48]. Adiponectin demonstrates the opposite effect. Its level rises in the blood in proportion to the reduction of body mass [49,50], a phenomenon that was also confirmed in the present study. Adiponectin can influence body mass reduction through the stimulation of the release of free fatty acids and glucose from peripheral tissues [51].

The present study did not show any changes in the blood concentration of IGF-1 in any of the groups. This is a rather unfavorable phenomenon because IGF-1 can promote muscle lipids and glucose metabolism and reduce insulin resistance [20,52]. Fontana et al. [22] in a study performed on men and woman showed that a 2-year calorie restriction diet caused a significant decrease in the concentration of IGF-1.

The former athletes participating in this study had a significantly increased body fat mass, which varied from 28% to 30% in most cases. Similar findings were reported by other authors [1]. Overweight and obesity among former athletes has been observed more often in athletes participating in speed–strength sport disciplines and those with weight categories in comparison to aerobic endurance sports [5,53,54]. This association is strongly related to body mass and fat mass during their sports careers [55,56,57,58]. Among NFL and NHL players, shot putters, and wrestlers in the heavyweight division, body mass helps in achieving better sport results. Borchers et al. [59] found that the prevalence of obesity was 21% in former NFL players. Kujala et al. [60] also confirmed that former power-sports athletes had a higher body mass compared with endurance athletes, especially long-distance runners and cyclists. Similarly, Albuquerque et al. [4] and Hyman et al. [54] showed that obesity is common in retired NFL players. 

Not all former athletes have problems with body mass. Former endurance-trained athletes have a much smaller likelihood of obesity [53,56]. Marquet et al. [53] confirmed that former cyclists have a significantly lower prevalence of obesity than the general population [53]. This may be caused by previously increased energy expenditure due to significant training loads, as well as high total daily energy expenditure compared with sedentary subjects [61]. In addition, Vogt et al. [62] indicated that elite cyclists had a 30% higher daily energy expenditure during the competitive season compared to preseason training. Other authors also explain this phenomenon by pointing to the very active lifestyle of retired endurance athletes [3,53]. Cyclists or marathoners even after the end of their careers maintain proper body weight and optimal body fat [49]. Phil and Jurimae [4] indicate that more than half of the former athletes in Finland engage in regular leisure-time physical activity or compete in different kinds of sports throughout their adult life, maintaining normal body weight. Similar observations were reported by Sarna et al. [2] in former Estonian athletes. 

It can be concluded that calorie-restriction diets are effective in reducing body mass in overweight and obese former athletes. Reducing daily caloric intake by 30% of TDEE is especially suitable for subjects with high body fat content, as well as increased glucose, insulin, lipoprotein and cholesterol levels. The reduction of body mass by 3 kg or more allows for a significant improvement in the lipid profile, leading to favorable endocrine changes. The concentration of leptin is reduced while that of adiponectin likely increases, stimulating the release of free fatty acids and glucose from peripheral tissues. It seems that former athletes should adopt a dietary strategy which allows for the preservation of muscle mass while reducing body fat content. Reducing daily caloric intake by 30% of TDEE can be recommended for overweight and obese former athletes to maintain proper body mass and health.

The present study has several strengths but also some limitations. The former includes a unique, large research group, the use of two calorically-different diet models, and a relatively long 6-week dietary intervention. The main limitation is the adherence of the study participants to the prescribed diet, effective control of dietary intake by the subjects, and the inability to unify the groups by preparing the same variable-calorie meals at the same time of day. These variables likely affected the final results of weight reduction, which are often quite divergent within the same dietary group.

## 5. Conclusions

For overweight and obese former athletes, a calorie restriction diet based on a 30% reduction of TDEE is more effective with regards to improved body mass, lipid profile, and reduced insulin and HOMA-IR levels compared to a 20% calorie restriction diet. Both the 20% and 30% of TDEE restriction diets reduced the levels of leptin and increased adiponectin concentration. 

## Figures and Tables

**Figure 1 nutrients-11-01461-f001:**
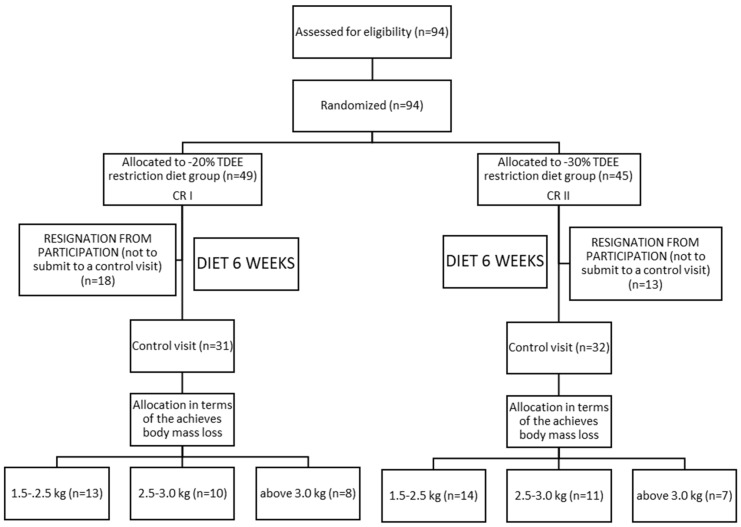
Scheme of the experimental protocol.

**Table 1 nutrients-11-01461-t001:** Characteristics of the applied diets.

Nutrients	CR I MEAN ± SD	CR II MEAN ± SD
TEI, kJ	9589.73 ± 150.6	8924.48 ± 217.6
TEI, kcal	2292 ± 36	2133 ± 52
CARBOHYDRATES, %	50 ± 0.4	50 ± 0.2
CARBOHYDRATES, g/kg/body mass	2.7 ± 0.3	2.6 ± 0.5
SIMPLE SUGARS, %	<10	<10
FIBER, g/day	33 ± 0.2	34 ± 0.5
FAT, %	30 ± 0.6	30 ± 0.8
FAT, g/kg body mass	0.8 ± 0.2	0.75 ± 0.3
CHOLESTEROL, mg/day	300 ± 2.1	300 ± 1.7
PROTEIN, %	20 ± 0.7	20 ± 0.06
PROTEIN, g/kg body mass	1.2 ± 0.6	1.1 ± 0.4

Note: TEI—Total Energy Intake.

**Table 2 nutrients-11-01461-t002:** The anthropometrics characteristic of the participant at the baseline.

Variable	CRI, *n* = 31 Mean ± SD	CR II, *n* = 32 Mean ± SD
BM, kg	92.3 ± 11.5	89.5 ± 14.0
BMI	28.1 ± 4.2	28.3 ± 4.1
BF, %	30.4 ± 8.0	29.3 ± 8.2
FFM, %	62.84 ± 8.36	58.98 ± 16.34
FFM, kg	67.68 ± 4.77	63.47 ± 17.48
MM, %	49.13 ± 4.59	43.36 ± 11.34
TBW, %	45.24 ± 5.26	42.35 ± 11.57

Note: CR—calorie restriction, BM—body mass; BMI—body mass index, BF—body fat, FFM—free fat mass, MM—muscle mass, TBW—total body water.

**Table 3 nutrients-11-01461-t003:** Results of Shapiro-Wilk tests for CR I and CR II.

Variables	Shapiro-Wilk Test	
	CR I	CR II
BM, kg	0.756	0.767
BMI	0.687	0.707
BF, %	0.686	0.715
FFM, %	0.713	0.743
FFM, kg	0.730	0.760
MM, %	0.749	0.778
TBW, %	0.764	0.791
GL, mmol/L	0.792	0.817
TCh, mg/dL	0.805	0.828
HDL, mg/dL	0.814	0.837
LDL, mg/dL	0.825	0.846
TG, mg/dL	0.835	0.855
TL, mg/dL	0.844	0.863
I, mg/dL	0.851	0.869
HOMA-IR	0.858	0.874
Leptin, ng/mL	0.791	0.813
Adiponectin, ng/mL	0.815	0.833
IGF-1, ng/mL	0.818	0.841

**Table 4 nutrients-11-01461-t004:** Differences in body composition, lipid profile and glucose control variables before and after CRI and CRII diet.

Variable		CR I (*n* = 31)			CR II (*n* = 32)			
		Before	After	ES	Before	After	ES	ES CR II vs. CR I
		Mean ± SD	Mean ± SD		Mean ± SD	Mean ± SD		
Body composition	BM, kg	92.3 ± 11.5	90.4 ± 10.7	0.014	89.5 ± 14.0	87.1 ± 12.6	0.031	0.037
	BMI	28.1 ± 4.2	27.8 ± 3.7	0.011	28.3 ± 4.1	27.3 ± 4.0	0.010	0.011
	BF, %	30.4 ± 8.0	29.2 ± 5.8	0.013	30.71 ± 5.9	29.0 ± 7.7	0.027	0.014
	FFM, kg	62.84 ± 8.36	58.95 ± 7.26	0.015	59.98 ± 16.34	58.02 ± 14.82	0.024	0.012
	FFM, %	67.68 ± 4.77	67.62 ± 4.95	0.010	66.39 ± 14.10	63.47 ± 17.48	0.055	0.045
	MM, %	49.13 ± 4.59	48.70 ± 6.07	0.011	48.39 ± 12.62	43.36 ± 11.34	0.058	0.051
	TBW, kg	45.24 ± 5.26	45.18 ± 4.83	0.010	43.95 ± 11.80	42.35 ± 11.57	0.026	0.034
Lipid profile	tCh, mg/dL	197.2 ± 40.3	171.4 ± 53.4 *	0.343 *	198.1 ± 40.5	187.3 ± 37.1 #*	0.241	0.351 #*
	HDL, mg/dL	51.6 ± 13.4	47.0 ± 16.2	0.061	54.7 ± 13.3	56.4 ± 17.9	0.021	0.061
	LDL, mg/dL	107.6 ± 28.3	102.5 ± 64.7	0.072	122.8 ± 42.0	116.3 ± 34.8 #*	0.253	0.347 #*
	TG, mg/dL	102.3 ± 57.3	89.7 ± 54.8 *	0.333	108.5 ± 45.8	83.0 ± 48.7 *	0.249 *	0.057
	TL, mg/dL	599.51 ± 102.7	520.80 ± 105.1 *	0.349 *	630.44 ± 124.4	572.93 ± 97.3 *	0.135 *	0.355
Glucose control	GL, mmol/L	5.24 ± 0.83	4.86 ± 1.27	0.070	5.34 ± 1.53	5.17 ± 1.01	0.013	0.067
	I, mg/dL	10.9 ± 6.9	8.7 ± 5.2	0.034	10.0 ± 8.2	9.06 ± 5.8	0.021	0.028
	HOMA-IR	2.6 ± 2.3	2.4 ± 1.1	0.010	2.7 ± 1.3	2.0 ± 1.4	0.010	0.012
	Leptin, ng/mL	13.78 ± 4.8	9.45 ± 6.9 *	0.148 *	13.27 ± 4.7	9.69 ± 3.9 *	0.144 *	0.019
	Adiponectin, ng/mL	3.33 ± 1.3	7.10 ± 3.2 *	0.166 *	3.83 ± 1.2	6.93 ± 3.3 *	0.147 *	0.022
	IGF-1, ng/mL	43.93 ± 4.6	42.50 ± 14.8	0.041	44.76 ± 10.4	42.93 ± 6.6	0.023	0.011

Note: ES—effect size, BM—body mass; BMI—body mass index, BF—body fat, FFM—free fat mass, MM—muscle mass, TBW—total body water, TL—total lipids, TG—triglycerides, tCh—total cholesterol, HDL—cholesterol HDL, LDL—cholesterol LDL, GL—glucose, I—insulin, IGF-1—insulin-like growth factor, *—statistically significant difference with *p* < 0.01 compared with before and ES for after, #—statistically significant difference with *p* < 0.05 compared with the CRI and ES for CRII; #*—statistically significant difference with *p* < 0.05 compared with the CRI and ES for CRII and after.

**Table 5 nutrients-11-01461-t005:** Results of the CR I and CR II diet on body mass and composition before and after the diet intervention.

**Variables**	**CR I**					
	**1.5–2.5 kg**		**2.5–3.0 kg**		**Over 3 kg**	
	**Before**	**After**	**Before**	**After**	**Before**	**After**
BM, Kg	97.95 ± 4.5	92.26 ± 3.1	99.60 ± 6.1	94.75 ± 4.2	105.67 ± 5.2	100.91 ± 8.1
BMI	31.26 ± 1.2	28.14 ± 0.2	30.65 ± 2.13	29.23 ± 1.3	32.19 ± 2.5	30.86 ± 2.1
BF, %	30.68 ± 3.1	29.92 ± 2.1	32.46 ± 5.3	31.85 ± 3.4	37.61 ± 3.4	37.10 ± 5.3
FFM, %	69.32 ± 4.2	68.05 ± 3.64	67.60 ± 4.2	66.54 ± 5.3	63.02 ± 4.2	62.39 ± 5.2
FFM, Kg	59.52 ± 3.1	55.75 ± 2.36 *	60.70 ± 4.1	55.91 ± 4.2	55.94 ± 4.1	53.53 ± 2.5
MM, %	49.97 ± 5.3	46.91 ± 2.77 *	49.56 ± 3.1	46.97 ± 2.7	42.96 ± 2.6	42.67 ± 2.9
TBW, %	50.87 ± 5.3	49.19 ± 3.36	49.61 ± 4.2	48.55 ± 3.1	47.59 ± 3.9	46.84 ± 2.8
**Variables**	**CR II**					
	**1.5–2.5 kg**		**2.5–3.0 kg**		**Over 3 kg**	
	**Before**	**After**	**Before**	**After**	**Before**	**After**
BM, Kg	96.95 ± 8.9	93.37 ± 6.3	100.43 ± 4.2	96.27 ± 4.3	98.67 ± 3.8	93.15 ± 2.1
BMI	30.08 ± 1.3	28.99 ± 1.4	30.78 ± 2.1	29.56 ± 1.1	30.40 ± 2.1	28.30 ± 1.4
BF, %	32.89 ± 2.1	31.86 ± 2.4	34.44 ± 2.6	33.75 ± 1.3	33.61 ± 2.1 #	31.21 ± 0.9 #
FFM, %	67.60 ± 3.1	65.84 ± 4.2	65.56 ± 4.2	65.21 ± 3.8	68.79 ± 4.9 #	67.90 ± 5.3
FFM, Kg	58.48 ± 4.1	57.86 ± 3.2	58.53 ± 3.6	55.98 ± 4.1	59.71 ± 3.2	54.86 ± 3.1
MM, %	47.83 ± 2.1	46.92 ± 3.5	46.83 ± 2.6	44.74 ± 2.5	49.86 ± 2.1 #	46.64 ± 2.5
TBW, %	50.25 ± 4.1	48.86 ± 3.4	48.85 ± 3.6	47.82 ± 4.1	50.59 ± 3.2 #	48.56 ± 3.5

Note: BM—body mass; BMI—body mass index, BF—body fat, FFM—free fat mass, MM—muscle mass, TBW—total body water, *—statistically significant difference with *p* < 0.05 compared with baseline, #—statistically significant difference with *p* < 0.05 compared with the CRI.

**Table 6 nutrients-11-01461-t006:** Results of the CR I and CR II diet on lipid profile and glucose control variables before and after the diet intervention in particular subgroups.

**Variables**		**CR I**					
		**1.5–2.5 kg**		**2.5–3.0 kg**		**Over 3 kg**	
		**Before**	**After**	**Before**	**After**	**Before**	**After**
Lipid profile	TL, mg/dL	735.63 ± 33.5	609.65 ± 41.23	775.00 ± 47.3	665.36 ± 63.1	608.86 ± 51.3	528.50 ± 72.1
	TG, mg/dL	131.63 ± 13.37	84.41 ± 14.91 *	127.30 ± 8.2	84.45 ± 13.1	126.00 ± 11.2	82.63 ± 8.2
	tCh, mg/dL	201.56 ± 35.26	162.35 ± 26.13	200.90 ± 26.1	166.55 ± 21.4	181.14 ± 38.5	165.00 ± 27.2
	LDL, mg/dL	132.25 ± 14.81	104.47 ± 12.1	139.90 ± 12.5	110.45 ± 9.2	120.86 ± 17.1	99.87 ± 15.5
	HDL, mg/dL	49.81 ± 2.94	49.12 ± 3.2	46.37 ± 5.3	47.62 ± 8.2	49.43 ± 3.2	50.07 ± 4.1
Glucose control	GL, mmol/L	5.30 ± 1.42	4.78 ± 0.6	5.50 ± 0.8	4.95 ± 1.7	5.51 ± 1.5	5.12 ± 1.2
	I, mg/dL	11.08 ± 2.62	10.14 ± 1.3	13.52 ± 4.1	12.86 ± 3.1	11.79 ± 3.7	10.18 ± 4.2
	HOMA-IR	2.68 ± 0.31	2.43 ± 0.24	3.36 ± 1.3	3.15 ± 1.5	2.82 ± 0.8	2.36 ± 0.7
	Leptin, ng/mL	12.32 ± 1.2	9.76 ± 1.3 *	13.10 ± 1.4	8.45 ± 0.4 *	15.43 ± 2.3	9.19 ± 1.3 *
	Adiponectin, ng/mL	3.51 ± 0.2	6.45 ± 0.5 *	3.50 ± 0.3	7.55 ± 0.5 *	3.12 ± 0.6	7.20 ± 1.2 *
	IGF-1, ng/mL	42.23 ± 0.9	41.90 ± 2.1	43.06 ± 1.1	41.60 ± 0.8	44.21 ± 3.4	43.10 ± 1.5
**Variables**		**CR II**					
		**1.5–2.5 kg**		**2.5–3.0 kg**		**Over 3 kg**	
		**Before**	**After**	**Before**	**After**	**Before**	**After**
Lipid profile	TL, mg/dL	639.71 ± 42.3	559.36 ± 23.2 *	632.70 ± 54.2	540.40 ± 21.1 *	649.14 ± 32.1	548.57 ± 24.5 *
	TG, mg/dL	99.50 ± 4.2	71.36 ± 12.1	109.00 ± 12.3	76.70 ± 21.5	109.00 ± 15.1	73.86 ± 23.6
	tCh, mg/dL	204.54 ± 18.1	183.64 ± 13.1 *	199.40 ± 14.2	172.80 ± 11.2 *	206.71 ± 14.1 #	177.57 ± 12.2 *
	HDL, mg/dL	55.50 ± 8.2	56.63 ± 6.1	50.80 ± 3.1	49.89 ± 4.1	50.42 ± 4.1	50.76 ± 3.4
	LDL, mg/dL	131.43 ± 9.4	112.64 ± 7.3 *	126.90 ± 8.2	107.60 ± 8.2 *	134.5 ± 7.27	112.29 ± 9.4 *
Glucose control	GL, mmol/L	5.07 ± 1.1	5.04 ± 0.8	5.34 ± 0.7	5.12 ± 0.9	5.39 ± 0.8	5.10 ± 1.1
	I, mg/dL	10.77 ± 1.8	8.34 ± 1.3	12.29 ± 1.3	8.72 ± 2.8	14.99 ± 2.1	9.90 ± 3.1
	HOMA-IR	2.52 ± 0.5	1.93 ± 0.9	2.91 ± 0.7	2.13 ± 0.5	3.57 ± 1.4	2.46 ± 1.1
	Leptin, ng/mL	11.86 ± 2.1	8.13 * ± 2.5	12.10 ± 3.1	9.65 * ± 0.3	15.15 ± 1.1	10.30 * ± 1.2
	Adiponectin, ng/mL	4.08 ± 1.2	8.10 * ± 2.6	3.84 ± 3.2	7.20 * ± 0.4	3.60 ± 0.8	6.95 * ± 0.4
	IGF-1, ng/mL	44.35 ± 2.7	43.16 ± 3.1	42.94 ± 2.9	43.08 ± 1.5	42.40 ± 1.9	41.30 ± 0.8

Note: TL—total lipids, TG—triglycerides, tCh—total cholesterol, HDL—cholesterol HDL, LDL—cholesterol LDL, GL—glucose, I—insulin, IGF-1—insulin-like growth factor, *—statistically significant difference with *p* < 0.05 compared with baseline, #—statistically significant difference with *p* < 0.05 compared with the CRI.

## Data Availability

The datasets used and/or analyzed during the current study are available from the corresponding author upon reasonable request.

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
