# Peer review of "Six Weeks of Calorie Restriction Improves Body Composition and Lipid Profile in Obese and Overweight Former Athletes"

_nutrients, 2019, doi:10.3390/nu11071461_

Round 1
Reviewer 1 Report
The authors conducted a 6-week calorie restriction trial for obese and overweight former athletes and concluded that a 30% reduction of TDEE was superior in losing body weight and in improving lipid profile and insulin resistance to a 20% reduction of TDEE. It is interesting to focus on the health issue of retired athletes who are often suffering from weight gain. While to clarify the effect of calorie restriction on their anthropometrical factors as well as lipid and glucose metabolism is meaningful, there remains some concerns as described below.
The improvement of lipid profile and insulin resistance were not observed in the participants in the CRI group who have equally achieved a body weight loss of 3 kg or more. How do the authors explain this result? Does the strictness of the calorie restriction itself have more impact on the improvement of lipid profile and insulin resistance than the extent of body weight loss? The point needs to be fully discussed.
It is very hard to believe that the participants in the CRI group with small extent of body weight loss have shown a marked increase in FFM and MM even though their physical activity level remained unchanged throughout the intervention. Furthermore, the change in their FFM (%) was minor while they achieved a body weight loss of > 5 kg in average (Table 2). Did they really gain FFM and MM? If so, dividing into 3 subgroups according to the extent of body weight loss has a different implication between CRI and CRII groups. The main body component which was lost following the intervention would have been different (fat or muscle) between the groups. Instead of that, would it be better to divide into subgroups according to the extent of body fat loss? At least, it needs to be further discussed how the dividing simply according to the extent of body weight loss could affect the results.
The completion rates of both interventions were not necessarily high notwithstanding the authors’ statement, “mild caloric restriction.” The authors need to add some comments about how do they think about this low completion rate?
The authors concluded that insulin levels were normalized, and insulin resistance was reduced in the participants in the CRII group. The expression would be too much even though the changes were not statistically significant.
Author Response
Dear Reviewer,
Thank you for your time and the detailed review of our manuscript. Your
valuable comments and critiques have improved the manuscript. All comments
were considered and included in the revised manuscript. In the text below, we have tried to respond to all of the reviewers comments.
The authors conducted a 6-week calorie restriction trial for obese and overweight former athletes and concluded that a 30% reduction of TDEE was superior in losing body weight and in improving lipid profile and insulin resistance to a 20% reduction of TDEE. It is interesting to focus on the health issue of retired athletes who are often suffering from weight gain. While to clarify the effect of calorie restriction on their anthropometrical factors as well as lipid and glucose metabolism is meaningful, there remains some concerns as described below.
Point 1. The improvement of lipid profile and insulin resistance were not observed in the participants in the CRI group who have equally achieved a body weight loss of 3 kg or more. How do the authors explain this result? Does the strictness of the calorie restriction itself have more impact on the improvement of lipid profile and insulin resistance than the extent of body weight loss? The point needs to be fully discussed.
Response 1: Losing weight and increasing physical activity independently has a beneficial effect on the glucose metabolism and insulin sensitivity (Lin et al. 2009). The improvement in insulin sensitivity with dietary intervention depends on the number of calories reduced. The more restrictive diet the greater improvement in insulin sensitivity. Our study participants demonstrated no significant improvement as they consumed a relatively high number of calories daily (2192±51kcal). Insulin sensitivity improvement can be observed in healthy subjects even in short-lasting but very low-calorie diets, where calorie intake can only be 500kcal /d. Such a diet model induces markedly decreases in abdominal adipose tissue, reduction in adipocyte size, an increased adipose tissue gene expression of mitochondrial biogenesis markers and non-mitochondrial oxygen consumption pathways, as well as improved whole-body insulin sensitivity (Vink et al., 2017)
Lin CY, Chen PC, Kuo HK, Lin LY, Lin JW, Hwang JJ. Effects of obesity, physical activity, and cardiorespiratory fitness on blood pressure, inflammation, and insulin resistance in the National Health and Nutrition Survey 1999-2002. Nutr. Metab Cardiovasc Dis2009, 20, 713-719.
Vink RG, Roumans NJ, Čajlaković M, Cleutjens JPM, Boekschoten MV, Fazelzadeh P, Vogel MAA, Blaak EE, Mariman EC, van Baak MA, Goossens GH. Diet-induced weight loss decreases adipose tissue oxygen tension with parallel changes in adipose tissue phenotype and insulin sensitivity in overweight humans. Int J Obes (Lond). 2017, 41(5), 722-728. doi: 10.1038/ijo.2017.38. Epub 2017 Feb 9.
This issue is interesting thus we decided to add this part to the discussion.
Point 2. It is very hard to believe that the participants in the CRI group with small extent of body weight loss have shown a marked increase in FFM and MM even though their physical activity level remained unchanged throughout the intervention. Furthermore, the change in their FFM (%) was minor while they achieved a body weight loss of > 5 kg in average (Table 2). Did they really gain FFM and MM? If so, dividing into 3 subgroups according to the extent of body weight loss has a different implication between CRI and CRII groups. The main body component which was lost following the intervention would have been different (fat or muscle) between the groups. Instead of that, would it be better to divide into subgroups according to the extent of body fat loss? At least, it needs to be further discussed how the dividing simply according to the extent of body weight loss could affect the results.
Response 2: As the reviewer correctly observed, marked increases in FFM and MM after the reduction diet are quite controversial. After a detailed analysis of raw results and the statistics file it appeared that while putting the results into the table a mistake was made regarding the data presented as “before” and “after” the intervention. Mixed reduction diets quite often result in both muscle and fat tissues being reduced. Reduction in fat tissue is a far more biochemically complex process which is regulated by hormones. It is believed that in order to reduce fat tissue but at the same time maintain a high level of muscle tissue, more protein should be included in the diet, approx. 1.6-1.8 g/kg of BM (Michalczyk et al., 2019). With regard to CRI and CRII the protein intake was approx. 1 g/kg of BM, which value is recommended in a mixed diet for adults. Due to the comparable level of muscle and fat tissue reduction in both the CRI and CRII groups, we have decided to leave the original subdivision into 3 subgroups according to the reduced number of body mass.
Michalczyk M, Chycki J, Zajac A, Maszczyk A, Zydek G, Langfort J. Anaerobic Performance after a Low-Carbohydrate Diet (LCD) Followed by 7 Days of Carbohydrate Loading in Male Basketball Players. Nutrients 2019, 11(4), 778; https://doi.org/10.3390/nu11040778
Point 3: The completion rates of both interventions were not necessarily high notwithstanding the authors’ statement, “mild caloric restriction.” The authors need to add some comments about how do they think about this low completion rate?
Response 3: This issue is interesting thus we decided to add this part to the discussion.
Point 4: The authors concluded that insulin levels were normalized, and insulin resistance was reduced in the participants in the CRII group. The expression would be too much even though the changes were not statistically significant.
Response 5: This expression was changed

Reviewer 2 Report
Summary: The aim of this study is to compare the effects of two (20% or 30%) calorie restricting diets over six weeks on body composition and metabolic profile among overweight and obese former male athletes. This study should be of interest to a broad population and has implications beyond both male and athletic populations.
Broad comments: Strengths of this study include a moderate sample size with numerous physiologic measures.
The main weakness of this is presentation of the results. It’s unclear why the results were split into weight loss categories or if pairwise comparisons were made. Considering the reduction in group sample size by splitting two groups into 9 groups, effect size analysis is recommended. Presentation of results is confusing, at best.
Comments: First person should not be used. Mixed font sizes occur randomly throughout, correct for consistency. Both of these issues are mentioned below, but these are not all cases of the problems
Title and Keywords
Abstract
General: Objective should be stated as an objective and first person removed. There are many acronyms making both methods and results hard to follow, consider grouping related items (eg. body composition, glucose control, lipid profile). Summary of results would be clearer by how variables are significantly different (eg. what were the significant differences in TG, LDL, and TL (lipid profile)?)
Line 28: correct typo “6”, written number conventions should be used (eg. 6 should be six
Introduction
General: The introduction presents all the key variables are these are related to the primary aim of the study and loosely related to the population of the interest. Paragraphs 4 and 5 (IGF-1 and HOMA-IR) would be stronger if literature was presented related these variables to the population of interest as opposed to an explanation of the significance of the test itself.
Lines 74-75: mixed font sizes, correct
Materials and Methods
General:
Line 125: removed end parentheses or add open parentheses as needed
Lines 134-135: Indicate if subgrouping by weight loss was a priori.
Lines 157-160: explain the Fitmate apparatus as indirect calorimetry and what measure from this was used for the TDEE calculation. Explain how the activity factor was determined-what question(s) were asked?
Lines 160-165: this appears to be results
Line 180: explain how the nutritionist validated the diet records
Lines 212-213: citation needed for HOMA-IR formula
Statistical analysis
General: For significantly different baseline measures consider ANCOVA; also consider calculating effect size for p<0.05 to express the clinical meaningfulness of the results
Results
General: This section should lead with a summary of the results based on the aim of the study. Referring to tables is not beneficial to the reader. State how results were significantly different, eg. After 6 weeks the CRI diet resulted in a significant increase(?) in FFM (kg) and MM (kg). The first paragraph broad summary of results is hard to follow and unclear if significant differences are for each diet as a whole or by weight loss. The analyses by weight loss categories should be clearly explained and presented. Overall presentation of results is poorly described and organized.
Discussion
General: Considering the high attrition in this study, consider addressing this limitation in the discussion and recommendation of 30% CR
Lines 318-319: statement that it’s especially beneficial for those with glucose and lipid disorders is not appropriate, these were specifically exclusion criteria for this population
Conclusions
General: Generally well-written. After fixing results section, this may need some rewriting
Table 1: as formatted, this is very hard to read. CRI and CRII are based on TDEE-clarify the values given the table, eg. are PROTEIN g/kg body mass the average of all subjects who completed?
Table 2: This is results, presented in methods. Mixed boldface font; use footnote criteria according to journal guidelines
Table 3: this is baseline data and should be appear at the beginning of results or earlier, before Table 2; it also appears to be the same as Table 2, just reorganized by weight lost. Use footnote criteria according to journal guidelines
Tables 4 and 5: Like tables 2 and 3, these appear to present the same data.
Figure 1: Figure requires a title; this is a CONSORT diagram, not study design (Lines 170-171)
Author Response
Dear Reviewer,
Thank you for your time and the detailed review of our manuscript. Your
valuable comments and critiques have improved the manuscript. All comments
were considered and included in the revised manuscript. In the text below, we have tried to respond to all of the reviewer's comments.
Point 1: The main weakness of this is presentation of the results. It’s unclear why the results were split into weight loss categories or if pairwise comparisons were made. Considering the reduction in group sample size by splitting two groups into 9 groups, effect size analysis is recommended. Presentation of results is confusing, at best?
Response 1:
The presentation of the results was completely rearranged. Two new tables were added, i.e. Tables 2 and 3. The previous version of Table 2 was changed, while the previous Tables 3 and 5, as the reviewer suggested, were removed. Currently, Table 3 includes results of all groups following CR I and CR II, before and after the diet intervention. Since data presented in the new Table 3 did not demonstrate any statistically significant changes before and after the intervention considering intragroup and intergroup comparisons, we decided to keep the division into 3 subgroups in CRI and CRII, according to the participants’ body mass reduction.
Point 2: First person should not be used. Mixed font sizes occur randomly throughout, correct for consistency. Both of these issues are mentioned below, but these are not all cases of the problems
Response 2:
Done as requested
Point 3: Abstract
General: Objective should be stated as an objective and first person removed. There are many acronyms making both methods and results hard to follow, consider grouping related items (eg. body composition, glucose control, lipid profile). Summary of results would be clearer by how variables are significantly different (eg. what were the significant differences in TG, LDL, and TL (lipid profile)?)
Line 28: correct typo “6”, written number conventions should be used (eg. 6 should be six
Response 3:
Done as requested
Line 28 –corrected
Point 4: Introduction
General: The introduction presents all the key variables are these are related to the primary aim of the study and loosely related to the population of the interest. Paragraphs 4 and 5 (IGF-1 and HOMA-IR) would be stronger if literature was presented related these variables to the population of interest as opposed to an explanation of the significance of the test itself.
Lines 74-75: mixed font sizes, correct
Response 4:
Additional literature about the HOMA- IR was included. However, there is no literature related to the IGF- 1 in the population of interest. Our study is the first one conducted in this research area.
Line 74-75- done as requested
Point 5: Materials and Methods
Point General:
Line 125: removed end parentheses or add open parentheses as needed
Lines 134-135: Indicate if subgrouping by weight loss was a priori.
Lines 157-160: explain the Fitmate apparatus as indirect calorimetry and what measure from this was used for the TDEE calculation. Explain how the activity factor was determined-what question(s) were asked?
Lines 160-165: this appears to be results
Line 180: explain how the nutritionist validated the diet records
Lines 212-213: citation needed for HOMA-IR formula
Statistical analysis General: For significantly different baseline measures consider ANCOVA; also consider calculating effect size for p<0.05 to express the clinical meaningfulness of the results
Response 5:
Line 125- The use of parentheses was revised.
Lines 134-135- After conducting the experiment, the study participants were further divided into 3 subgroups depending on the range of body mass reduction. This part is clearly explained in the material and methods part in the section regarding subjects.
Line 157-160 – The precise COSMED Fitmate system was used to determine the RMR energy expenditure. The whole test took approximately 20 minutes. The device uses the indirect calorimetry method and directly measures the amount of oxygen uptake (with accuracy of ± 0.02%), which makes it possible to measure the resting energy expenditure (REE), the resting metabolic rate (RMR) and the basic metabolism rate (BMR). The Canopy measuring cup was used for the measurement, which is a scientific standard in this procedure. In order to establish the activity factor (AF), each participant during the first visit was administered a validated International Physical Activity Questionnaire - IPAQ. This questionnaire is divided into 5 parts and refers to work-related PA as well as activities performed at home and its surroundings, moving around, activities performed in free time (in relation to moderate or intensive physical activity). The last questions concern the time spent sitting. The reviewer’s suggestion regarding explanation of this part is very valuable, thus we decided to add this part to the dietary intervention part.
Line 165-165- After a minor correction we believe this part is suitable for methods section.
Line 180- Diets were reconstructed from dietary diary entries received from the participants in the Diet 5 program and referred to the currently valid standards recommended by the National Food and Nutrition Institute in Warsaw. This issue is interesting thus we decided to add this part to the Diet control part.
Line 212-213- Done as requested
Statistical analysis
We appreciate the reviewer's suggestions, although we have decided to use ANOVA because it is an effective technique for carrying out research in various disciplines such as business, economics,
psychology, biology and education when there are one or more samples involved. It is often confused with ANCOVA, as both are used to check the variance in the mean values of the dependent variable
associated as a result of controlled independent variables, after considering the consequences of the
uncontrolled independent variable. ANOVA is used to compare and contrast the means of two or
more populations. ANCOVA is used to compare one variable in two or more populations while
considering other variables. ANCOVA is an ANOVA model that has a general linear model with
a continuous outcome variable (quantitative, scaled) and two or more predictor variables, where
at least one is continuous and at least one is categorical (nominal, non-scaled). To sum up, in our study
ANOVA provides more precise results and does not narrow the output data from the model.
Effect size was calculated. The table with results was added (Table 3).
Point 6: Results
General: This section should lead with a summary of the results based on the aim of the study. Referring to tables is not beneficial to the reader. State how results were significantly different, eg. After 6 weeks the CRI diet resulted in a significant increase(?) in FFM (kg) and MM (kg). The first paragraph broad summary of results is hard to follow and unclear if significant differences are for each diet as a whole or by weight loss. The analyses by weight loss categories should be clearly explained and presented. Overall presentation of results is poorly described and organized.
Response 6:
As suggested by the reviewer, the table layout was changed and a new table (Tab 2) containing output data for the entire groups CRI and CRII was included. Tables comparing the results of intragroup comparison were removed and significant differences between the groups were shown in tables 4 and 5. As the reviewer correctly observed, FFM and MM results are controversial. After a detailed analysis of raw results and the statistics file it appeared that while putting the results into the table a mistake was made in the data presented as “before” and “after” the intervention regarding FFM and MM results. Due to the comparable level of muscle and fat tissue reduction in both the CRI and CRII groups, we have decided to leave the original subdivision into 3 subgroups according to the reduced number of body mass. We hope that the changes we brought into the results section will satisfy the reviewer. Should the reviewer be unhappy about the changes in this section they should not hesitate to contact us. We will gladly amend whatever the reviewer may suggest. Let us just say that we highly value Nutirents journal and we would really appreciate out study being published in its special issue.
Point 7: Discussion
General: Considering the high attrition in this study, consider addressing this limitation in the discussion and recommendation of 30% CR
Lines 318-319: statement that it’s especially beneficial for those with glucose and lipid disorders is not
appropriate, these were specifically exclusion criteria for this population
Response 7:
Done as requested
Line 318-319. Changed as requested
Point 8: Conclusions
General: Generally well-written. After fixing results section, this may need some rewriting
Response 8: Done as requested
Point 9 Tables and figure
Table 1: as formatted, this is very hard to read. CRI and CRII are based on TDEE-clarify the values given the table, eg. are PROTEIN g/kg body mass the average of all subjects who completed?
Table 2: This is results, presented in methods. Mixed boldface font; use footnote criteria according to journal guidelines
Table 3: this is baseline data and should be appear at the beginning of results or earlier, before Table 2; it also appears to be the same as Table 2, just reorganized by weight lost. Use footnote criteria according to journal guidelines
Tables 4 and 5: Like tables 2 and 3, these appear to present the same data.
Figure 1: Figure requires a title; this is a CONSORT diagram, not study design (Lines 170-171)
Respons 9:
Tab 1- Protein content presented in table 1 was defined at the beginning of the experiment for all subjects who started the experiment.
Tab 2- After major changes we decided to leave data in Tab 2 in the results.
Table 3 – We deleted Table 3
Table 4 and 5- We removed Table 5 and changed Table 4
Figure 1- The title was changed –“ Scheme of the experimental protocol”

Round 2
Reviewer 1 Report
The authors reliably addressed the comments.